Relationships among cost, citation, and access in journal publishing by an ecology and evolutionary biology department at a U.S. university

Peterson A. Townsend town@ku.edu 1
Cobos Marlon E. 1
Sikes Ben 1
Soberon Jorge 1
Osorio-Olvera Luis 2
Bolick Josh 3
Emmett Ada 3
1 Department of Ecology and Evolutionary Biology, University of Kansas , Lawrence , KS , USA
2 Departamento de Ecología de la Biodiversidad, Universidad Nacional Autónoma de México , Mexico City , CDMX , Mexico
3 KU Libraries, University of Kansas , Lawrence , KS , USA
Costello Mark
Electronic publication date: 2024 Jan 4
Publication date: 2024
Volume: 12
Electronic Location ID: e16514
Received 2022 May 6; Accepted 2023 Nov 2
Copyright: ©2024 Peterson et al.
Copyright year: 2024
Copyright holder: Peterson et al.
License: This is an open access article distributed under the terms of the Creative Commons Attribution License, which permits unrestricted use, distribution, reproduction and adaptation in any medium and for any purpose provided that it is properly attributed. For attribution, the original author(s), title, publication source (PeerJ) and either DOI or URL of the article must be cited.
License URL: https://creativecommons.org/licenses/by/4.0/

Keywords: Journals, Publication, Subscription, Article processing charges, Open access

Funding: The authors received no funding for this work.

==============================
Background

Optimizing access to high-quality scientific journals has become an important priority for academic departments, including the ability to read the scientific literature and the ability to afford to publish papers in those journals. In this contribution, we assess the question of whether institutional investment in scientific journals aligns with the journals where researchers send their papers for publication, and where they serve as unpaid reviewers and editors.

Methods

We assembled a unique suite of information about the publishing habits of our Department of Ecology and Evolutionary Biology, including summaries of 3,540 journal publications by 35 faculty members. These data include economic costs of journals to institutions and to authors, benefits to authors in terms of journal prestige and citation rates, and considerations of ease of reading access for individuals both inside and outside the university. This dataset included data on institutional costs, including subscription pricing (rarely visible to scholars), and “investment” by scholars in supporting journals, such as time spent as editors and reviewers.

Results

Our results highlighted the complex set of relationships between these factors, and showed that institutional costs often do not match well with payoffs in terms of benefits to researchers (e.g., citation rate, prestige of journal, ease of access). Overall, we advocate for greater cost-benefit transparency to help compare different journals and different journal business models; such transparency would help both researchers and their institutions in investing wisely the limited resources available to academics.

Introduction

Recent decades have seen major transformations in scholarly publishing practices, particularly in fields like ecology and evolutionary biology. Major recent milestones have included commercialization of most journals in the field, and consequent increases in subscription prices (McGuigan, 2008). In response, researchers at many institutions organized high-profile boycotts of particular publishing enterprises (Goveas, Sayer & Sud, 2022; Heyman, Moors & Storms, 2016), created rights-retention open access policies at the level of departments or institutions (Xia et al., 2012), and explored new publishing models (e.g., journal “membership”; Binfield, 2013; Else, 2018). More recently, many journals have shifted from subscription-based access, with publishing in most cases free to authors, to open access publishing with funding coming from article processing charges (APCs) to authors (Peterson et al., 2019b). These latter changes have indeed broadened reading access to journal-published papers around the world, but have simultaneously closed publishing access to potential authors who often cannot afford APCs (Larios et al., 2020; Mekonnen et al., 2022; Nabyonga-Orem et al., 2020; Peterson, Emmett & Greenberg, 2013).

These seismic changes in the scholarly publishing universe have led to a series of challenges for researchers. The first challenge was how to ensure that the full, worldwide community of researchers would be able to access (and cite) other researcher’s work, a challenge that gave rise to the open access movement. Even with expansions in open access, many university-based researchers have seen significant erosion of access to the subscription-based journals at their libraries: rising subscription costs have led most university libraries to cut subscriptions to increasing numbers of journals (SPARC, 2021). More recently, new challenges center on how our community of researchers can afford the APCs instituted by an increasing numbers of journals in the field to fund open access publication (Peterson, Emmett & Greenberg, 2013; Solomon & Björk, 2012).

Despite these sweeping changes in scholarly publishing and scholars’ roles in the scholarly publishing ecosystem, few or no analyses have examined publishing practices by researchers (but see Aczel, Szaszi & Holcombe, 2021), particularly as a reflection of how their institutions spend their limited resources. Here, we present a first analysis of the peer-reviewed, journal-based scholarly publishing practices of a single academic department—the Department of Ecology and Evolutionary Biology at the University of Kansas, which is a relatively large and reasonably productive department, ranked #66 among peer departments in a recent summary (Research.com, 2023). Although our analysis indeed covers only a single department, trends and relationships in the broader phenomenon of where researchers publish (and why) are of general importance across ecology and evolutionary biology and more broadly. Our goal is to characterize the journals in which faculty researchers publish their work, and analyze those choices in relation to the cost of particular journals (i.e., subscriptions, APCs), benefits to the researcher (e.g., citation rates for individual papers), Clarivate’s Journal Impact Factor™(JIF™), and the work and time that the researchers donate to the journals (e.g., by reviewing and editing).

Materials & Methods

Data compilation

The analyses presented in this paper are built upon a detailed compilation of data about publication, reviewing, and editing activities of department members, as well as data about the journals themselves. Our focus was on peer-reviewed, journal-published scholarly publications, not out of disregard or lack of appreciation for other forms of scholarly publication (e.g., monographs, books, book chapters, blog posts), but rather in view of the fact that, in ecology and evolutionary biology, peer-reviewed journal papers are the principal currency by which faculty are evaluated for promotion, tenure, and general achievement and advancement. We focused on journals in which University of Kansas Ecology and Evolutionary Biology (henceforth “EEB”) faculty (all faculty members combined) had published at least twice (i.e., we eliminated journals in which EEB faculty had published only a single paper).

EEB faculty publications

We derived a complete catalog of papers published by EEB faculty members during their careers. We first searched for individual “scholar profiles” on Google Scholar (https://scholar.google.com/) for each of the 38 then-current EEB faculty members (9 December 2022; no effort was made to include past EEB faculty in this search). Six EEB faculty members did not have Google Scholar profiles at the time of our original derivation of data for this study, in 2020, so we requested that they create profiles. We interviewed a number of EEB faculty members about their maintenance of their Google Scholar profiles, and none indicated any filtering or biasing behavior regarding including papers in their profiles; the only maintenance activity that was mentioned by any of the faculty members was that of “synonymizing” multiple versions of papers that had entered in the profile. In the end, 35 of 38 EEB faculty members had profiles; two of the three individuals who did not were not highly research-active. Journals represented at least twice among the pool of journal publications across all of these profiles were the basis for all our data analyses.

We used customized scripts written in R (R Core Team, 2020) and the package “scholar” (Keirstead, 2016) to “scrape” publication lists for each EEB faculty member (see EEB faculty scholar IDs and scripts available in https://doi.org/10.17161/1808.32587). We isolated from these data records the year, journal name, and total number of citations for each paper. For each individual publication, we calculated the average number of citations per year since publication as [total citations / (2023—publication year)]. We restricted our analyses to journals that are peer-reviewed, to the best of our knowledge, although we were unable to separate out publications in peer-reviewed journals that were nonetheless not peer-reviewed (e.g., book reviews).

EEB faculty reviewing and editing activities

EEB faculty members are required to submit annual reports to the department, which include information about their service activities. Certain data elements are extracted from these reports by department personnel, and are made available to the broader community for general information without identification of individual faculty members. From this information, we tallied the total number of manuscript reviews provided to each journal by EEB faculty over the period 2015–2018. From these same annual reports, we also derived a summary of average number of EEB faculty editor-years for each journal during the period 2015–2018.

Journal characteristics

For clarity in reference to journals, we added ISSN codes for each journal in which EEB faculty published papers via consultation of http://portal.issn.org and https://www.ncbi.nlm.nih.gov/nlmcatalog. When multiple ISSNs were available for a journal, we used the newer version or the online version. Finally, when doubt existed owing to multiple journals with similar or identical names, we searched the actual title of paper on Google Scholar, accessed the paper online, and then identified the journal that published the article(s) in question.

We next summarized the characteristics of each journal in multiple dimensions, via queries to a series of information sources. JIF™ numbers were obtained for 2019 via queries to Web of Science (https://wos-journal.info/). Whether journals were open access or not was obtained from the Directory of Open Access Journals (DOAJ) 16 December 2022 public data dump (https://doaj.org/docs/public-data-dump/). Note that our focus is on fully open publication of articles, such that the final published version is what is available; as such, we do not emphasize so-called “green” open access solutions, in which the author leverages their ability to post pre-publication copies of papers online, often in institutional repositories. For open access journals, we obtained costs of publication in the form of article processing charges (APCs) from the same source, in a few cases supplemented via direct consultation of journal web pages. Open access journals that do not charge APCs (i.e., “platinum” open access journals) were counted as having an APC of $0. We also evaluated journal friendliness to open access via the SHERPA/RoMEO site (https://v2.sherpa.ac.uk/romeo/), recording journals as “yes” (i.e., can post immediately) “wait” (i.e., can post, but after an embargo period), “$” (i.e., only via paying an open-access fee), “funder” (i.e., only under a funder mandate), or “no” (i.e., cannot post) for posting final published version, author’s last version(i.e., post-review), or the submitted version (i.e., submitted version). For this latter information, we assumed that open-access journals in which authors retain copyright, or in which content is served under a CC-BY license, would be “yes” under all three categories.

Finally, only for the purpose of illustration, we obtained what we term a “cost of openness” for each journal. This quantity is the APC in the case of open access journals, but is the hybrid open access fee in the case of subscription journals. The APC was obtained as described above, whereas the hybrid fee was obtained via direct consultation of journal web pages. We note that the potential for self-archiving of papers published in many journals (i.e., so-called “green” open access) is not considered in this “cost”: rather, we focus on the costs associated with full and open access to the final, published version of each paper.

Journal characteristics at the University of Kansas

We obtained information on the 2019 subscription price (paid annually) by means of searches of publisher websites, journal by journal. For the remaining journals, KU Libraries personnel were able to check journal subscription costs from internal sources. Note that this price information refers to subscriptions to closed-access journals only, such that we do not include this information for open access journals. Finally, via publishers’ COUNTER Code of Practice, Release 4, data provided to KU Libraries, we obtained data information on journal usage (access counts) during 2018–2022; COUNTER is an information source used by publishers to inform libraries about their journal usage.

Data harmonization, improvement, and analysis

We next invested considerable time and effort in standardizing journal names, to ensure that information from individual journals was not duplicated across name variations. Although Google Scholar records presented some journal naming issues (e.g., spelling, duplications), these complications were much more challenging when harmonizing data among sources (e.g., Google Scholar, Web of Science, DOAJ, SHERPA/RoMEO, EEB annual reports, etc.).

The final dataset was then subjected to a series of exploratory analyses, mainly bivariate plots and regression analyses, designed to detect patterns. All statistical analyses were carried out in R. The full, final dataset is available via the University of Kansas’s digital repository, KU Scholarworks, at https://doi.org/10.17161/1808.32708.

Results

Basic characteristics

In all, the 35 EEB faculty members published a total of 5,916 items that were listed in their Google Scholar profiles. Those profiles, however, included many items that were not peer-reviewed publications (e.g., abstracts), not journal publications (e.g., book chapters), or that duplicated other entries in profiles. As such, when we quality-controlled and cleaned the dataset, the total number reduced to 3540 journal publications.

Summarizing the data by journal instead of by faculty member, the number of EEB publications in journals that we analyzed ranged from 2 (see above) to 81 (Molecular Phylogenetics and Evolution). Other journals seeing frequent EEB publication included Zootaxa (76), Journal of the Kansas Entomological Society (76), PLOS One and ZooKeys (74), and Evolution (61) (Fig. 1). The journals with the fewest EEB publications were either journals in other fields, in which EEB researchers were often co-authors as part of early research experiences or out-of-field collaborations, or small and regional journals in which EEB faculty have published only occasionally.

Figure 1 Number of articles published, numbers of reviews, and average editorships per year, for different journals by ecology and evolutionary biology faculty.

Total number of reviews (2015–2018) and average number of editorships per year (2015–2018) are shown as color ramps from 0 (white) to maximum value (red). Numbers of papers in the same set of journals are shown in the histogram, excepting journals with <20 papers published by the group of faculty under analysis.

KU EEB investment in and contribution to journals

Over a recent 4-year period, EEB faculty contributed over 110 reviews to single journals (Evolution, Trends in Ecology and Evolution) (Fig. 1), but far fewer in other journals where they nonetheless publish their research frequently (e.g., PLOS ONE, Proceedings of the Royal Society B). EEB faculty editorships were focused in journals of modest publication activity (e.g., Systematic Biology, Zootaxa, Oecologia)—that is, some EEB faculty dedicate significant editorial time to some journals in which they do not often publish. Although both reviewing activity and editorships related positively and significantly to number of EEB papers published (Table 1), researchers’ investment of time as reviewers or editors was not related directly to annual subscription price either (Fig. 2).

Costs and benefits

A first consideration is the relationship between various measures of journal quality and journal price—i.e., are academic libraries and (less directly) university-based scholars getting their money’s worth when they pay for expensive journals? The relationship between JIF™ and annual subscription price showed a relatively strong, positive relationship (R2 = 0.220, P < 0.05). However, the positive slope of this relationship was a consequence only of Science and Nature, statistical outliers for JIF™ and both relatively high as far as price. Removing these two journals reduced the R2 by 83.1% to 0.0372 (Fig. 3). The relationship between JIF™ and numbers of EEB publications was positive, but not strong: the overall regression had R2 = 0.0236; removing two outlier journals (Science and Nature) reduced the strength of the relationship, to R2 = 0.0036; in only the former case was the relationship statistically significant (Fig. 3).

Table 1 Summary of relationships between independent (columns) and dependent (rows) variables in analyses of relationships among variables related to publishing by ecology and evolutionary biology faculty.

For each cell in the matrix, “ +” indicates positive-slope relationships, and “ −” indicates negative-slope relationships. Boldface indicates relationships for which the slope is significantly different from zero, and the R2 value is given in parentheses in all cases.

	Number of EEB papers	2019 subscription price	Article processing charges (US$)	Cost of openness (US$)	Average EEB editorships per year	Total EEB reviews 2015–2018	Journal Impact Factor™	Average citations per year	Average usage (2018–2022)	
Number of EEB papers		+ (0.0021)	− (0.0031)	+ (0.0005)	+ (0.1669)	+ (0.1042)	+ (0.0236)	+ (0.0132)	+ (0.0684)	
		+ (0.0004)	− (0.0031)	+ (0.0005)	+ (0.1734)	+ (0.1044)	+ (0.0036)	+ (0.0082)	+ (0.0489)	
2019 subscription price	+ (0.0021)		+ (0.0331)	+ (0.1237)	− (0.0018)	+ (0.0002)	+ (0.2198)	+ (0.0429)	+ (0.1054)	
	+ (0.0004)		+ (0.0331)	+ (0.1237)	− (0)	+ (0.0017)	+ (0.0372)	+ (0.0167)	− (0)	
Article processing charges (US$)	− (0.0031)	+ (0.0331)		+ (0.1033)	− (0.0437)	+ (0.0028)	+ (0.6793)	+ (0.1083)	+ (0.0612)	
	− (0.0031)	+ (0.0331)		+ (0.1033)	− (0.0437)	+ (0.0028)	+ (0.6793)	+ (0.1083)	+ (0.0612)	
Cost of openness (US$)	+ (0.0005)	+ (0.1237)	+ (0.1033)		− (0.0082)	+ (0.0021)	+ (0.0351)	+ (0.0141)	− (0.0017)	
	+ (0.0005)	+ (0.1237)	+ (0.1033)		− (0.0082)	+ (0.0021)	+ (0.0351)	+ (0.0141)	− (0.0017)	
Average EEB editorships per year	+ (0.1669)	− (0.0018)	− (0.0437)	− (0.0082)		+ (0.009)	− (0.0029)	+ (0.0001)	+ (0.0021)	
	+ (0.1734)	− (0)	− (0.0437)	− (0.0082)		+ (0.009)	− (0.0031)	+ (0.0001)	+ (0.0114)	
Total EEB reviews 2015–2018	+ (0.1042)	+ (0.0002)	+ (0.0028)	+ (0.0021)	+ (0.009)		+ (0.0122)	+ (0.0264)	+ (0.0016)	
	+ (0.1044)	+ (0.0017)	+ (0.0028)	+ (0.0021)	+ (0.009)		+ (0.0235)	+ (0.0259)	+ (0.0017)	
Journal Impact Factor™	+ (0.0236)	+ (0.2198)	+ (0.6793)	+ (0.0351)	− (0.0029)	+ (0.0122)		+ (0.1453)	+ (0.4772)	
	+ (0.0036)	+ (0.0372)	+ (0.6793)	+ (0.0351)	− (0.0031)	+ (0.0235)		+ (0.1583)	+ (0.0674)	
Average citations per year	+ (0.0132)	+ (0.0429)	+ (0.1083)	+ (0.0141)	+ (0.0001)	+ (0.0264)	+ (0.1453)		+ (0.0581)	
	+ (0.0082)	+ (0.0167)	+ (0.1083)	+ (0.0141)	+ (0.0001)	+ (0.0259)	+ (0.1583)		+ (0.0271)	
Average usage (2018–2022)	+ (0.0684)	+ (0.1054)	+ (0.0612)	− (0.0017)	+ (0.0021)	+ (0.0016)	+ (0.4772)	+ (0.0581)		
	+ (0.0489)	− (0)	+ (0.0612)	− (0.0017)	+ (0.0114)	+ (0.0017)	+ (0.0674)	+ (0.0271)		

Figure 2 Two metrics of faculty investment in journals (numbers of reviews and editorships), as a function of 2019 subscription price.

Blue trendline shows a simple linear regression including all journals, whereas the red trendline shows the relationship when the outlier journals Science and Nature are excluded from the analysis. Journals falling well away from the main cloud of journals (i.e., additional outliers) are labeled for discussion in the text.

Figure 3 Number of articles published, plus three metrics of payoff to faculty for publishing in journals (Journal Impact Factor™, average citations per year, average usage in the University of Kansas Libraries system), as a function of 2019 subscription.

Blue trendline shows a simple linear regression including all journals, whereas the red trendline shows the relationship when the outlier journals Science and Nature are excluded from the analysis. Journals falling well away from the main cloud of journals (i.e., additional outliers) are labeled for discussion in the text.

Journal usage at the University of Kansas showed a positive relationship to annual subscription price (R2 = 0.1054, P < 0.05). Again, however, the relationship was a consequence of the inclusion of Science and Nature and not a more general trend: removing those two journals eliminated the positive slope of the relationship and reduced the R2 dramatically, effectively to 0, and the relationship was no longer statistically significant (Fig. 3, Table 1). Benefits to researchers in terms of citation rate, however, showed no relationship to annual subscription price. Overall, the relationship was weak (R2 = 0.0429, P > 0.05; Fig. 3). Articles published in Science and Nature did not show any marked elevation in citation rate compared to other journals, such that the relationship did not change with their inclusion (R2 = 0.0167, P > 0.05; Fig. 3, Table 1).

For open access journals, EEB faculty published as many as 81 articles in single journals (PLOS ONE). The APCs for publishing in open access journals ranged from $0 (i.e., “platinum” open access journals) to $5,300 (PLoS Biology). EEB faculty did not tend to publish more in free journals or in expensive journals (Fig. 4), but rather these publication choices appeared to be unrelated to APCs (R2 = 0.003, P > 0.05).

Figure 4 Number of articles published and two metrics of payoff to faculty for publishing in open-access journals (Journal Impact Factor™, average citations per year), as a function of article processing charges (in US$).

Blue trendline shows a simple linear regression including all journals, whereas the red trendline shows the relationship when the outlier journals Science and Nature are excluded from the analysis. Journals falling well away from the main cloud of journals (i.e., additional outliers) are labeled for discussion in the text.

The relationship between JIF™ and APCs was decidedly positive, with few or no platinum open access journals being accorded JIF™ ratings (= indexing in Web of Science). Indeed, JIF™ increased by ∼2 for every $1,000 of increase in APCs (R2 = 0.679, P <0.05; Fig. 4, Table 1). Numbers of citations that individual papers received showed a positive relationship to APCs (R2 = 0.1083, P < 0.05; Fig. 4, Table 1).

Finally, the cost of openness in journals in which EEB faculty published showed some rather impressive sums. That is, for two journals (Nature Ecology & Evolution, Nature Plants), the cost of making a published paper open access was above $9,000. In another seven journals, costs of making a published paper open access were above $5,000. Simply for the purposes of illustration, the total cost of publication, the sum of numbers of papers multiplied by the cost of openness for each journal where EEB faculty published their work is above $900,000.

Discussion

We are not aware of datasets or analyses similar to what we present here, although one recent study calculated the value of peer review that is donated by scientists to the publishing industry (Aczel, Szaszi & Holcombe, 2021). That is, we have assembled a suite of novel pieces of information, including where academics publish their work; how much those publications cost the academics (APCs), their institutions (e.g., institutional support for paying APCs, subscription charges), or their funders; how much those publications benefit the academics (e.g., in terms of citation rate or journal impact factor ratings, or download rates within the institution); and how academics choose to support journals via unpaid editing and reviewing activities. Assembling this information required scraping algorithms for harvesting data from Google Scholar, by-hand summary of data in annual reports and online databases, and careful consultation with University of Kansas Libraries personnel as regards annual subscription prices and download rates. Exploration of this dataset therefore can inform about important, yet often unperceived, relationships.

Getting your money’s worth

To what degree are institutional investments of scarce financial resources “paying off” in the currencies of importance in academia, such as research impact, prestige, and citation rates? The extent to which impact factors are (or should be) of importance in judging academic excellence is hotly contested (Anonymous, 2016; Saha, Saint & Christakis, 2003; Seglen, 1997). Our analyses revealed that, without the outliers Nature and Science, three metrics that may be seen as assessing payoff or benefit to researchers (i.e., JIF™, citation rate, journal usage by readers), show no relationship to subscription costs—that is, expensive journals do not “pay off” in greater return to the researcher. Nature and Science rank among the most expensive of all KU EEB-related journals and represent outliers in several of our analyses. Similarly, those two journals (plus Scientific Reports and Proceedings of the National Academy of Science USA) are used at KU much more than the usage-price relationship among the remaining journals (Fig. 3). Undoubtedly, this effect might be attributed to the multi-disciplinary nature of these journals, whereas most of the rest are limited to ecology and evolutionary research dimensions. With or without those few, highest-profile journals, the relationship between number of publications by EEB faculty and impact factor was positive, but quite weak and not statistically significant.

Curiously, though, when one focuses on numbers of citations accrued by individual publications (“citation rates”) instead of journal-level impact factors, relationships change dramatically (Neylon & Wu, 2009), both for subscription-based and open access journals. Overall, citation rates per year were only slightly positively related to subscription costs. Among subscription-based journals, the standout journals were Ecography, Ecological Modelling, and Nature, each with modest subscription prices, or sufficiently high numbers of citations that a higher subscription price was possible. Similarly, among open access journals, citation winners were Ecography (in transition from subscription-based to open access), Science Advances, Royal Society: Open Science, and Nature Communications. Note that this calculation omits the so-called platinum open access journals, which have no APCs—journals with highest citation rates in this category include Emerging Infectious Diseases, Revista Mexicana de la Biodiversidad, Journal of Librarianship and Scholarly Communication, Revista da Sociedade Brasileira de Medicina Tropical, and Biodiversity Informatics. These contrasts between citation rates and impact factors as a basis for valuing journals have been pointed out in other contexts (Galligan & Dyas-Correia, 2013; Vanclay, 2009), but suggest that the current shift toward article-based valuation metrics may lead to consequent shifts in emphasis on different sets of journals as “winners”.

Journal Impact Factor (IF™) is a proprietary, journal-level metric calculated and published annually by Clarivate Analytics (Clarivate, 2021). In brief, this index is a proprietary and not-reproducible function of the mean number of citations in a given year of citable articles over the previous two years. Although impact factors were devised by a librarian and bibliometrician, who intended it primarily as a tool for use by librarians in collection development and management (Garfield, 1972), they have since been taken up by a commercial enterprise. JIF™ has been the subject of numerous critiques of both its inherent nature and its (mis)use (Baldwin, 2017; Juyal et al., 2019; McVeigh & Mann, 2009; PLOS Medicine Editors, 2006). Garfield himself stated “that you should not use JIF to evaluate a person or department” (Kim, 2000), a sentiment that was underlined in 2012 with the signing of the San Francisco Declaration on Research Assessment (DORA) (DORA, 2012). DORA was subsequently signed by >24,000 individuals and organizations in 164 countries (as of Dec. 2023).

Article impact is more appropriately assessed through article-level metrics, such as numbers of citations or citation rates, as well as by a number of so-called “alternative” metrics (“alt-metrics”), such as views and downloads, social media attention, and news reporting on an article (Altmetric, 2021). Of course, the best assessment of the quality of an article is to read the article, but this can quickly become unwieldy for evaluators such as those on hiring, promotion, and funding committees. Article-level metrics also have limitations and must be used appropriately, but they do address the article rather than the venue of publication, and therefore represent a more suitable set of tools than JIF™ or other journal-level metrics of any sort for assessing the quality, impact, and merit of individual publications. These ideas are underlined by the results of this study, in which the “best-value” journals for researchers and academic institutions are nuanced and subtle in the qualities that distinguish them.

Investing time and energy in the right journals

One can imagine diverse motivations for scholars to invest their time in particular journals: keeping current with the literature, supporting a particular field, allegiance to a scientific society, and the prestige of reviewing or holding an editorial position at a high-profile journal. KU EEB faculty reviewed manuscripts with particular frequency for journals such as Evolution and Trends in Ecology and Evolution, in which they published frequently, but also for prestigious journals like Current Biology, where KU EEB publication is not as frequent. Similarly, KU EEB editing effort was focused in journals such as Systematic Biology, Zootaxa, and Oecologia, which were not particularly high in frequency as publishing venues for KU EEB faculty.

KU EEB “investment” in journals was also not related to price. One might expect that such a relationship might be positive, if price were a good indicator of value or prestige, or that it would be negative, if KU EEB faculty were investing their time and effort in journals that are good bargains for them and the university. In fact, the relationship was not at all clear, such that we suspect that numerous other priorities and motivations direct how faculty dedicate their time and effort. This set of motivations would seem to represent an interesting and fruitful area for future research.

Perspectives for the future

The landscape of scholarly publishing is continuing its rapid shift. Older, subscription-based models are clearly decreasing in number, with journal after journal “going open” (e.g., Araújo, Svenning & Tuomisto, 2019), notwithstanding the mixture of good and bad consequences from those changes (Peterson et al., 2019a). This shift mirrors the earlier, massive-scale commercial investment in academic publishing and its effects. Once again, commercial interests (i.e., profit, market share, prestige) frequently outweigh the academic motivations of communication and access, leading to discords and mismatches between the publishing and academic communities and their respective interests (Buranyi, 2017).

Journal-level metrics are heavily used by committees conducting job searches or evaluation junior faculty for promotion and tenure. Our data show that, at least for the case of KU EEB, the most expensive journals are not necessarily those most cited or even, excluding a few outliers, those with highest impact. Our results clearly highlight that, for individual researchers, investment in service to certain prestigious and expensive journals makes dubious sense. What to do then? The idea that impact factors should be used to assess individual productivity has been so strongly criticized (Schimanski & Alperin, 2018; Vanclay, 2012) that we wonder why it continues being used. Better ways to evaluate academic productivity have already been suggested, particularly in the form of article-level quality metrics (Schimanski & Alperin, 2018).

A consequence of these evolving publishing modes and outcomes has been the broad exploration of new models of subscription and payment—in effect, business models for academic publishing. Significant experiments include the “Public Library of Science”, and its exploration of not-for-profit open access journals (which nonetheless have high APCs) (Bernstein et al., 2003); another novel business model is the author membership-based PeerJ (which also ends up charging the equivalent of a significant APC in the absence of an institutional membership) (Binfield, 2013). Another force in this academic publishing world has been that of open access mandates, particularly by funding agencies (e.g., Suber, 2008), and so-called “green” open access, in which individual researchers make versions of their research open and available. This opening of access absent financial return to the journals, regardless of annual subscription prices or APCs, was originally feared to bankrupt academic journals; however, in the case of the largest funder mandate to date in the United States, it did not result in noticeable changes in viability and survival of journals in the fields that were most affected (Peterson et al., 2019b). Indeed, in some cases, changes forced on journals by commercial publishing enterprises have led to mass resignations of editorial boards, exemplifying the priority contrasts between the different stakeholders in the academic publishing world (Peterson et al., 2019a).

Conclusions

Academics play a curious role in a world with many stakeholders and interests: academics produce the journal publications that are the currency of that world, and need access to the totality of journal publications for both publishing and reading. Nonetheless, those same academics are subject to changing university budgets and commercial interests that affect the publishing world directly. This paper has summarized the outcome of myriad influences and interests, in terms of how they translate into publishing patterns, for one Department of Ecology and Evolutionary Biology, in this case at the University of Kansas. The degree to which these patterns are unique to this department versus more broadly representative will paint an intriguing picture of how scholars publish in the 21st century.

Supplemental Information

Supplemental Information 1 Data

Data on journals in which University of Kansas Ecology and Evolutionary Biology faculty have published 2 or more papers. See data table key.

Click here for additional data file.

Supplemental Information 2 Key

Definitions, source, and remarks on each field in the data table.

Click here for additional data file.

This article would not have been possible without the collaboration of the staff of the University of Kansas Libraries, particularly Kristin Sederstrom, Debbra Peres, Heather Mac Bean, and Angie Rathmel. Thanks also to Dorothy Johanning for assistance with summarizing relevant data from EEB annual reports.

Additional Information and Declarations

Competing Interests

Author Contributions

Data Availability

The authors declare there are no competing interests.

A. Townsend Peterson conceived and designed the experiments, performed the experiments, analyzed the data, prepared figures and/or tables, authored or reviewed drafts of the article, and approved the final draft.

Marlon E. Cobos performed the experiments, analyzed the data, authored or reviewed drafts of the article, and approved the final draft.

Ben Sikes conceived and designed the experiments, authored or reviewed drafts of the article, and approved the final draft.

Jorge Soberon conceived and designed the experiments, authored or reviewed drafts of the article, and approved the final draft.

Luis Osorio-Olvera performed the experiments, analyzed the data, prepared figures and/or tables, authored or reviewed drafts of the article, and approved the final draft.

Josh Bolick conceived and designed the experiments, performed the experiments, authored or reviewed drafts of the article, and approved the final draft.

Ada Emmett conceived and designed the experiments, authored or reviewed drafts of the article, and approved the final draft.

The following information was supplied regarding data availability:

The scripts for obtention of data are available at KU ScholarWorks: https://doi.org/10.17161/1808.32587.

The data are available at KU ScholarWorks: https://doi.org/10.17161/1808.32708.

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
