# Peer review of "Relationships among cost, citation, and access in journal publishing by an ecology and evolutionary biology department at a U.S. university"

_PeerJ, doi:10.7717/peerj.16514_

## Round 0.1 · original submission · Major Revisions

Of the 3 reviews, while all are positive one delved deeper into the underlying data and found a number of errors which are a concern. This must be addressed in a revision, and as it should not be expected that the referee checked every example, then all the journal's OA policy and APC should be rechecked. All the referees make some additional constructive suggestions as well.

·

Basic reporting

I have suggested some additional references that could be added in the introduction and Discussion sections; also a slight edit of Figure 3a.
Have made some suggestions - see document attached - to reword parts of the text for greater readability.

Experimental design

I have suggested some additional information to be provided regarding the statistical analyses performed.
Have also asked some minor points to be clarified in the methods - see attache document.

Validity of the findings

Have made some suggestions of additional discussion points - see attached document.

Additional comments

This is a well-written and well-structured study that will be of interest to the whole of the scientific community, especially environmental scientists.
I have made some minor comments / suggestions throughout the text, to provide more clarity / detail and improve readability.
Great effort by the authors.

·

Excellent Review

This review has been rated excellent by staff (in the top 15% of reviews)
EDITOR COMMENT
This review was exceptionally good in that the reviewer checked some of the assumptions and data underpinning the analysis, noticed some probable errors (perhaps made my assistants coding the data), and made constructive comments on how the authors could improve the paper.

Basic reporting

In this manuscript the authors gather and analyze publication data on 34 out of 39 (over 87%) of the faculty of the Department of Ecology & Evolutionary Biology, at the University of Kansas (KU), as of March 1st 2020, to examine in which venues they publish, edit, and review-for, and the costs and prices associated with each of those publication venues.

I think this is really interesting work. I’m glad to see that knowledgeable KU librarians were brought into the collaboration to help with the work - similar research done by academics-only such as https://peerj.com/articles/7850/ has faced criticism and doubts over validity for not including librarians.

Key to this manuscript are the data, the software, and the exact processing/coding steps. I thank the authors for publicly archiving this in a suitable repository so they can be available for the reviewers and the public.

With regards to the underlying data https://kuscholarworks.ku.edu/handle/1808/32708 did the authors upload the correct file?

The data requires major revisions to make it robust, accurate, and clear.

In terms of coding “OA friendly” - I think a lot of journals have been miscoded, or else the coding methodology is insufficiently described to encapsulate what was done.

Take for instance the journal Nature , the authors score this as ‘0’ in terms of “OA friendly”. From the manuscript I presume ‘0’ translates to “unfriendly (i.e. author not allowed to post any post-review version of their paper)”. Assigning ‘0’ to Nature seems factually untrue to me. In actual fact, the journal Nature permits authors to make public their post-review accepted manuscript after only a 6-month embargo. See https://www.nature.com/nature-portfolio/editorial-policies/self-archiving-and-license-to-publish “When an article is accepted for publication in a Nature Portfolio journal via the subscription route, authors are permitted to self-archive the accepted manuscript on their own personal website and/or in their funder or institutional repositories, for public release six months after publication”

Likewise, the journal Evolution, is given a '0' in the author's dataset, but it permits authors to upload their post-review accepted manuscript after a 12-month embargo (according to Sherpa/Romeo). I think the authors need to have a look again at how they’ve scored each of these journals for “OA-friendly” and perhaps even consider a more sophisticated coding scheme to encompass the variation in length of embargo and other OA-related attributes? The coding scheme of 0,1,2 is arbitrary and unusual, it needs more justification at the very least.

Why are there unfilled/blank cells and what do unfilled cells mean for each column?
For instance, I don’t list all <blank> cells, just emblematic examples which need to be addressed:

a) the journal Molecular Ecology has a KU 2019 Subscription Price of <blank>. Molecular Ecology is NOT a fully OA journal, so the value in this cell ought to be >0 and not <blank>. If KU does not currently subscribe to Molecular Ecology then state that in the cell. Even then, I would have thought a representative price quote could still be sought and entered? Needs explanation at the very least!

b) Average (COUNTER) Usage of 2016 of PLOS ONE is given as <blank>. I think this ought to be coded more explicitly as e.g. UNKNOWN or DATA UNAVAILABLE rather than <blank>

c) In their dataset, the journal Evolution has an Article Processing Charge value of <blank> . Evolution is a ‘hybrid’ journal that permits authors to have their article open access if they pay an APC, the 'standard' APC at 2022-05-17 is listed at a price of $3000 USD. <blank> insufficiently describes the situation of this journal in respect to providing open access &/or APC pricing. Please always be sure to refer to APCs as prices and not costs throughout the manuscript. The publisher often sets the APC as a price they determine, and this price may be completely divorced from the actual cost of running the publication venue.

d) Why is there no COUNTER usage data for Phil Trans of the Royal Society B? It is another unexplained <blank> where I would expect there to be >0 data given this is a subscription/hybrid journal and the publisher does seem at first glance to provide COUNTER usage data.

e) Why is “OA friendly” <blank> for many journals e.g. Phil Trans of the Royal Society B? This is just pure error - there must be a categorization for each and every journal given the current coding scheme.

f) Please provide an ISSN or ISSN-L for each journal, the journal name alone is not a unique identifier. “Journal of Insect Science” may refer to either one of the following two journals:
Journal of Insect Science (Entomological Society of America)
OR
Journal of Insect Science (Indian Society for the Advancement of Insect Science)

g) Not all open access journals are listed in DOAJ. Actually a great number of open access journals exist beyond DOAJ’s listings e.g. read the The OA Diamond Journals Study - Science Europe https://scienceeurope.org/our-resources/oa-diamond-journals-study/

This is more than just a theoretical problem. It is a practical one e.g. Bruce Liebermann has an article published in The Open Paleontology Journal (Bentham, now discontinued). This journal is/was fully open access. At one point in time The Open Paleontology Journal was DOAJ-listed, but the journal is no longer DOAJ-listed. Which brings me to another point: what version and/or timepoint of the DOAJ-listings did you use in this analysis? The DOAJ-listings are not static. It would be good to include some versioning information for reproducibility. If any journals were manually determined to be fully open access journals (outside of DOAJ-listings) perhaps state that? Pakistan Journal of Biological Sciences is one such title inc in the scope of this EEB analysis that to my eyes (looking at the journal website) seems openly accessible to readers, with articles distributed under CC-BY, yet it is not DOAJ-listed. Oddly "Pakistan Journal of Biological Sciences" is also only listed as "1" for "OA-Friendly" which must be an error given the final pdfs are open online under CC-BY?

I can't really trust this article until the underlying dataset is properly re-scored and more fully explained.

It would be nice if the authors wrote some nicer words about the value of research & scholarly writing that is NOT published in peer-reviewed journals. The way in which the 'not peer reviewed' or 'not published in a journal' were abruptly discarded from the analysis is troubling for the overall aim of the manuscript to look at what and where scholars and institutions are investing their time in supporting. Journals aren't the be all and end-all of scholarly communication. Monographs, book chapters, even blog posts can be excellent contributions to research, and all can be/are cited too.

Experimental design

What about selection bias? I note only 34/39 faculty were included and that the authors "requested they create [GS] profiles"
I think you should discuss the weakness of this design a bit further. How many of the faculty specifically created a GS profile at the authors request? Why would an EEB faculty member actively not want a GS profile and may people like this (anti-Google?) have a different publication, editor, and review profile?

What about the completeness of each of the 34 GS profiles included? No mention is made that GS profiles are malleable and owners of each profile can include/exclude publications on their GS profile at whim. I think you may want to consider evaluating this weakness, perhaps with reference to another data source?

Likewise, what is the completeness of annual reports submitted to the department detailing editorial handlings and peer review reports? If a faculty member was in a rush to submit their annual report, might they not accidentally forget to include an editorship or a peer review done here or there? Perhaps you can cross-check with Publons for one or more faculty members to help assure the reader of the completeness of this data?

Please include a link to an archived page as close as possible (in time) to the time-point at which you measured the number and identity of EEB faculty at KU (1st March 2020). I found a snapshot of the page at the Internet Archive from December 2019 here: https://web.archive.org/web/20191206051920/http://eeb.ku.edu/faculty but if you have any other/better/closer archived snapshots to prove that assertion then please use them.

Line 158 “high-ranking” high-ranking according to who or what? I disagree with the concept of ranking journals (see various works by e.g. Bjorn Brembs). When I do rank journals e.g. in a spreadsheet I tend to do it alphabetically, so American Naturalist is 'high-ranking' for me :)

The analysis fails to acknowledge explicitly that lots of peer-reviewed journals have specific sections within them that are NOT peer-reviewed. How was this accounted for, if at all? Thus an author may have published an item in Nature, but that item might not necessarily be peer-reviewed. Were 'editorials' , 'reviews', 'letters of concern' , 'erratum notices' , 'corrigenda' and other such exotic article types all counted the same as typical 'primary research articles' ? It might be good to remark on the effect of this if so in a 'limitations' section of this manuscript.

How representative is the EEB department at KU of any other EEB department at any other university? Some words to address this question might be nice?

Please consider more clearly indicating that Clarivate’s Journal Impact Factor (JIF)™ is a proprietary number. Specifically, capitalise all references to it AND include the ™ symbol to remind readers that it is also trademarked in addition to being untransparent, negotiable, statistically illiterate [1], irreproducible, and proprietary. Do not refer to merely ‘impact factor(s)’ – as such lower-casing might imply a generic, reproducible, statistically logical calculation. Also be more specific than just describing it as an 'average' (e.g. line 258). Clarivate famously calculates the mean rather than the median, which given the known highly skewed distribution of citations across articles in a journal is a statistically illiterate practice [1], hence you should tell readers it is the mean (which is statistically inappropriate) not the 'average'.

[1] Stephen Curry (2012) http://occamstypewriter.org/scurry/2012/08/13/sick-of-impact-factors/

Validity of the findings

The findings are not valid until they fix the dataset. The given dataset has a LOT of errors and unexplained cells.

Additional comments

Despite everything I've said above, I do think this manuscript presents some very interesting research and I would be keen to see it get published in a peer-reviewed journal. I merely wish to help make it more robust before it gets published in a peer-reviewed journal.

The calculus of investing time, labour, and hard cash in publication venues is an extremely important thing to examine and I congratulate the authors on gathering this data to start doing this, in the open so that others can see, think and learn from the results.

·

Basic reporting

Please see additional comments.

Experimental design

Please see additional comments.

Validity of the findings

Please see additional comments.

Additional comments

I was pleased to review “Relationships among cost, citation, and access in journal publishing by an ecology and evolutionary biology department at a U.S. university”. The topic of the paper is important and suitable for PeerJ. I have a few comments for improvement, which the authors should have no trouble addressing.

I was pleased to see that the authors archived the data and custom script used in this study. However, I would encourage the authors to also share the script used for the statistical analyses (test of relationships between variables) and to create the figures, as well as to improve the dataset by providing metadata (e.g., a comprehensive description of the column headings, units and abbreviations) in a separate readme file and applying tidy data principles (no blank cells, one value per cell [i.e., no units in cells]) to facilitate reuse (White et al. 2013).

Relationships between variables were examined by means of regression analysis, which assumes that one variable is dependent on the other. This dependency is not straightforward in almost all the relationships examined and, as such, the use of a correlative approach is more appropriate. The authors should consider reporting correlation coefficients rather than coefficients of determination. P-values should also be reported to three decimal places rather than above or below a critical value (Head et al. 2015).

The three paragraphs from L257-285 seem out of place and/or weekly linked to the text in this section of the manuscript. The content of these paragraphs is fine (although it could be condensed) but links to the discussion of the results are unclear. Consider rewriting, moving the text to another section, or drawing tighter links to the results discussed in the text of this section.

Numerous references cited in the text do not appear in the bibliography.

L55-64. Consider citing (Mekonnen et al. 2022) and (Larios et al. 2020).

L77. Consider defining ‘journal use’ on first use.

L79. Comparator needed in this sentence.

L172-174. This sentence seems more appropriate for the discussion.

L185. Consider specifying EEB publications at the University of Kansas.

L232. Consider referring the reader to the relevant figure.

Figure 3. The caption mentions publication frequency but the y axis in panel A is number of EEB papers. Please specify whether this is a frequency or a number and, in the latter case, the time period covered.

Regards,
Dom Roche

References

Head ML, Holman L, Lanfear R, Kahn AT, Jennions MD. 2015. The extent and consequences of p-hacking in science. PLoS Biology 13:e1002106.

Larios D, Brooks TM, Macfarlane NB, Roy S. 2020. Access to scientific literature by the conservation community. PeerJ 8:e9404.

Mekonnen A, et al. 2022. Can I afford to publish? A dilemma for African scholars. Ecology Letters 25:711-715.

White EP, Baldridge E, Brym ZT, Locey KJ, McGlinn DJ, Supp SR. 2013. Nine simple ways to make it easier to (re) use your data. Ideas in Ecology and Evolution 6:1–10.

---

## Round 0.2 · Minor Revisions

I only sent the paper to the more thorough referee. They are positive about the revised version but point out some possible errors or issues requiring checking or clarification. Please attend to them as appropriate.

I also suggest you rotate Fig 1 so the journal names are not all sideways and will thus be easier to read. Note the "Lorem ipsum" on Fig 2 which I assume is a mistake. Your results are very interesting and should cause reflection amongst scientists and taxpayer-funded institutions about the value they get from APC and Subscriptions.

·

Basic reporting

Much improved, thank you.

I will use this text box to respond to some comments made by the authors in the response to the first round reviews:

* The authors state that: "It appears that no members of the faculty were removing or biasing their [Google Scholar] profiles in any way that would affect the results of this study." does not strike me as particularly rigorous.

Just so that the authors understand the completeness problem here, I will laboriously explain it:

When logged-in to your Google Scholar profile, for any publication currently showing on your profile as one of your publications, you can click on it, then click “edit”. After clicking edit it takes you to screen where options are given including “Remove this article and its [XX] citations”. Similarly, if one has a common surname, Google Scholar can regularly give you an option to ‘claim authorship’ on a publication and have it listed in your Google Scholar profile as if you had authored it (even if you haven not!).

Google Scholar profiles therefore do not necessarily reflect the true publication profile of an individual as publications can be hidden/removed from Google Scholar profiles on whim AND in some cases it is also possible to _add_ publications to Google Scholar profiles that were not in fact authored by the person.

There is obvious incentive for people to delete publications they might consider 'embarrassing' from their public Google Scholar profile, and Google Scholar makes it very easy to do this. Asking a person face-to-face if they have ever selectively removed a publication from their Google Scholar profile is not an effective way of determining the truth (people lie!).

A spot check using an alternative independent (non-Google) data source on one or two scholars, using their name string and/or their ORCID might be a good way of reassuring readers that the Google Scholar profiles analyzed here are indeed a complete record of journal publishing events by those faculty, but this is merely a suggestion and I won't insist on it - it probably doesn't affect the results of this study, but would help to give readers the impression of rigor and awareness of the trust/completeness issues with Google Scholar profiles.

Experimental design

I thank the authors for a much improved manuscript and dataset.


Clarification: the "Average citations per year" column in the collected data.
Does this:

a) capture of the average citations (according to Google Scholar) for each and every paper in the journal
or
b) capture the average citations (according to Google Scholar) of _only_ Kansas EEB authored papers in the journal

I think it would be good to better document and clarify this unusual and non-standard measure.

``````

I think it is mis-framing to talk write in this manuscript about the 'cost of openness'. I think it more accurate to discuss the pricing of routes by which open access can be achieved. APCs are prices not costs. The publisher chooses a price to set, a price which may be entirely divorced from the basic costs of doing the work to the publisher.

Line 204: “Finally, only for the purpose of illustration, we obtained what we term a “cost of openness” for each journal. This quantity is the APC in the case of open access journals, but is the hybrid open access fee in the case of subscription journals.”

Although this route is not available at ‘pure’ open access journals, I must feel I should point out that most/all of the “hybrid” (paywalled) journals permit the green open access route (self-archiving). There is a cost of $0 associated with the green open access route. The analysis presented does not seem to factor-in the availability of green open access routes.
One of the authors listed on this manuscript (Josh Bolick) definitely knows the in’s and out’s of “green” open access and has published excellent work on it e.g. Bolick, J. (2017) Exploiting Elsevier’s Creative Commons License Requirement to Subvert Embargo. Kraemer Copyright Conference Poster Presentation. Colorado Springs, CO. June 5-6, 2017. I’m thus at a loss as to why green OA is so ignored in this manuscript until a very late and inconsequential mention at the end. If the central thesis is that hybrid APC pricing is appallingly high (and it is!) -- would it not be helpful to point out to colleagues that open access can also be achieved at a cost of $0 via green open access, and particularly via perhaps under-utilized methods such as the Bolick manoeuvre (Bolick, 2017).

Validity of the findings

I'm still a bit concerned about the validity of some individual statements in this manuscript and the data.

a) the seven journals named that according to this manuscript have charged Kansas EEB faculty over $9000 to make a single article open access within them.

I know that the list price for hybrid OA at Nature Ecology & Evolution, and Nature Plants is over $9000 - those I do not blink at. But for the other 5 journals named, the list prices given at the publisher site don't make it at all obvious how Kansas EEB faculty were charged over $9000 and so I think they need checking and or an explanation given (even if it is just an explanatory note in supplementary materials).

For Aquatic Insects (Taylor & Francis), if the paying authors are located in the United States, the list price for hybrid OA for all articles types (“research articles” “original paper” “method” and “book review”) is at this point in time $3500 https://authorservices.taylorandfrancis.com/choose-open/publishing-open-access/open-access-cost-finder/?category=all&journal=naqi&fulloa=0&openselect=1&notavailable=0&dove=1&routledge=1&tandf=1&numberofresultsperpage=5&pagenumber=1

For Annales de la Societe Entomologique de France (Taylor & Francis), for hybrid OA, if the paying authors are located in the United States, the list price for hybrid OA for all articles types is at this point in time $3500 https://authorservices.taylorandfrancis.com/choose-open/publishing-open-access/open-access-cost-finder/?category=all&journal=tase&fulloa=0&openselect=1&notavailable=0&dove=1&routledge=1&tandf=1&numberofresultsperpage=5&pagenumber=1

Aquatic Microbial Ecology is published by Inter-Research. For this journal the hybrid APC pricing information is clearly given on their website: https://www.int-res.com/journals/ame/about-the-journal/#tab2box
“Articles submitted on or after 1 November 2022: €2500 for Research or Review articles
Articles submitted on or after 1 June 2021 and before 1 November 2022: €2000 for Research or Review articles”

How did Kansas EEB end up paying over $9000 for open access for a single article in the journal Aquatic Microbial Ecology? I cross-checked with the openAPC database which has records of over 190,000 APC payments contributed from over 400 research institutions. No university has ever paid anything like that to Inter-Research for a single article, according to openAPC data. https://github.com/OpenAPC/openapc-de

Deutsche Entomologische Zeitschrift is published by Pensoft. Their pricing information is clearly explained here: https://dez.pensoft.net/about#Core-Charges
For an article that is 41-50 pages long the APC pricing is listed as € 1,400 with a charge of €15 euros for each additional page above that. By my calculations one would have to submit a manuscript with over 472 pages above the 50 page limit, in order to rack up a total bill in excess of $9000 for a single article in this journal. Whilst this is within the realms of possibility, it seems unlikely and perhaps deserves explaining to readers the exceptional case of a very lengthy article if that is indeed the cause of the high bill in this case.

These are very specific assertions about businesses and if they are factually incorrect assertions I don't doubt that the publishers of these journals would be very unhappy about it, so please for your own sakes double-check these ">$9000 for a single article" assertions.

b) I checked the new, revised dataset upload

• Acta Palaeontologica Polonica – this journal is listed in DOAJ and has been listed (without interruption) in DOAJ since 2005: https://doaj.org/toc/1732-2421

Yet in their dataset the authors have this journal scored in their “Is DOAJ OA?” column as (blank), which I presume is “No”. Why?

The categorization of journals as wholly open access (or not) seems very confused, but perhaps this does not matter to the specific analyses presented in the manuscript?

I picked out
* Acta Botanica Cubana
* Bulletin of Zoological Nomenclature [open access at the biotaxa platform but behind a paywall at the BioOne platform] https://www.biotaxa.org/bzn
* Truebia
all of these I would call open access journals.
Lots of the cells were coded with "not 100% clear!" which I feel is a bit amateurish.


The "2019 subscription price (US$)" column remains conspicuously empty / unfilled with data entered for only ~86 journals.

Subscription journals included in the dataset like Bioinformatics (published by OUP) do have subscription list prices available online e.g. at https://academic.oup.com/pages/purchasing/institutional-journals-products/subscribe-by-title#price-list for OUP and here for Wiley subscription journals: https://onlinelibrary.wiley.com/library-info/products/price-lists (e.g. for Zoologica Scripta, a Wiley journal where no subscription price is given in the dataset)
It is a little disappointing that the data gathering for this column appears haphazard. I appreciate that journals are bundled, and sometimes non-disclosure agreements have been signed, but even so, why not just use the public list prices?

Additional comments

line 531-532: "three metrics of benefit to researchers (i.e., JIF, citation rate, journal usage by readers)"

I think this needs re-wording. Do you really mean to imply that Journal Impact Factor is 'of benefit' to researchers? Perhaps you meant 'of interest ' ? I'm not sure that the regime of Journal Impact Factor mania necessarily benefits research or researchers (on the whole).

---

## Author Rebuttal · Round 0.2

To the Editor:

We have taken an extraordinarily long time to do this revision for a number of reasons. Principal among them was that we ended up redoing the dataset almost in its entirety (i.e., all except three data fields for which more up-to-date information was not available). Given that, we also redid all of the analyses, all of the figures, etc. Given the new analysis, as well as publication data from a number of new faculty members who joined the department, we feel that the analysis is now quite a bit more robust. The reviews that were provided to us were challenging, we will admit. Responding to each one of the many reviewer comments in the detail that it deserves was another reason why we were so tardy in providing this revision.

In sum, we very much appreciate the patience that you and the *PeerJ* staff have shown with us. We are confident that this revision is thorough, and that the manuscript is now much-improved over the original one.

Please do not hesitate to let us know if you have any questions or require any further information.

All the best,
Town Peterson, on behalf of the author team

**Reviewer: Paris Stefanoudis** (Note that we have drawn these comments from the document that the reviewer attached to their review)
- Line 45: Would be good if you could include a new reference as that one is slightly old.
  - *2022 reference added.*

- Line 56: Since it concerns the whole of the scientific community, I would reword to "for researchers." or "for the whole of the research community, including ecologists and evolutionary biologists". I prefer the former.
  - *Former suggestion accepted and added!*

- Line 65-66: on the role of scholars to the publishing ecosystem, you could mention here the value of scholars' reviewing that has been estimated to be worth billions of dollars… Aczel, B., Szaszi, B. and Holcombe, A.O., 2021. A billion-dollar donation: estimating the cost of researchers' time spent on peer review. Research Integrity and Peer Review, 6(1), pp.1-8.
  - *Thanks! We had not seen that paper previously. Now cited in the Introduction and the Discussion.*

- Lines 76-80: I think this could be removed as it pre-empts the results section.
  - *Now removed.*

- Lines 122-125: It would be good to link these three categories with established open access options such as green, gold etc., that are familiar to most readers. If I am not

mistaken that would be: 'fully friendly' = Gold, Diamond and Platinum; 'intermediate' = Green, and, 'unfriendly' = traditional subscription publishing model.

- *The linkages are not correct, as proposed by the reviewer. That is, there are traditional subscription-based publishing models that are friendly to OA, and there are "gold" OA journals that do not allow scholars to post copies of their own work. We have added verbiage in the Methods section to clarify to what we are referring in this section.*

- Lines 126-128. If they were subscription-based would they not be paid by the library as well? How are they different from the ones you got the information from the library? Adding that information would help readers.
  - *Yes, in theory at least. In reality, however, the Library is constrained by non-disclosure clauses in the subscription contracts that they handle. There was apparently a situation some years back in which university libraries compared subscription prices, and found massive discrepancies among then in terms of what they were being charged. As a consequence, the publishing companies have added clauses to the contracts to avoid libraries disclosing or sharing that information.*

    *Regardless, the KU Libraries staff was able to assist us in generating this information… The bulk of the information came from journal websites. The Library was able to supplement in some cases in which information was lacking on the journal websites and in which the language in their contracts for the subscriptions was permissive. We added text to clarify this point in the Methods section of the manuscript.*

- Line 133: Why not 2015-2018?
  - *The usage data have now been re-generated by the Library personnel. The data now cover the period 2018-2022.*

- Line 137: Better to reword to: "We standardized journal names, to ensure that…."
  - *Done.*

- Lines 143-147: Please provide more detail about the types of analyses conducted and only for those that were included in the final manuscript. If it's just bivariate plots and regressions, then reword to indicate that only these two analyses have been used to identify patterns in this study using R. On the latter, please also indicate the version of R used, and cite any statistical packages if applicable.
  - *Done.*

- Line 157: Maybe reword from "(see above)" to "(see Materials & Methods)"
  - *Done.*

- Line 159: Is that Proceedings of the Royal Society B: Biological Sciences? Must be, based on line 168 and Figure 1.
  - *Clarified.*

- Line 166: 'Over a recent 4-year period'. Best to specify the period (2015-2018).
  - *Done.*

- Line 166-167: '100+ reviews to single journals'. Best to replace with 'more than 100 reviews in two journals'.
  - *Done.*

- Lines 173-174. I would remove that sentence from the Results section, as it is more appropriate for the Discussion.
  - *Done.*

- Lines 177-179. Again, I would remove those lines as they way they are phrased are more for the Discussion. Or it could be included in the Methods section under the data analysis, and then mention the type of stats used to answer that question.
  - *We have rephrased these lines for clarity, but they are included in the Results section simply to frame why and how the section is structured the way it is. Again, rephrased to avoid it appearing to be Discussion.*

- Line 181. R2=0.211 is more relatively weak than relatively strong. If not comfortable with that, perhaps consider removing 'relatively strong'.
  - *Done.*

- Line 182. Consider rewording to: 'statistical outliers for both impact factor and price'.
  - *Done.*

- Line 195. Instead of weak I would focus on how the relationship was non-significant, and thus reword lines 193-195 to: "Benefits to researchers in terms of citation rate, however, showed no relationship to annual subscription (R2 = 0.038, P > 0.05; Figure 3)."
  - *Done.*

- Line 199-200. Consider rephrasing to: "The most popular open access journal was PLOS One, with a total of 31 publications by EEB faculty over the 2015-2018". Also it works better as the second sentence of the paragraph, right after the different price options are given.
  - *Done.*

- Lines 217-220. I would remove this sentence as it is more about the methodology. Also the novelty of the article is mention at the beginning of the discussion.
  - *Done.*

- Line 227: Anonymous, 2016. Is there no author for this article (could not find it when I searched for it) or could it be cited in a different way? Not sure, if it is wrong, but I have never seen Anonymous being used before. Alternatively, you can remove that citation and choose others, e.g. Harvey, L.A., 2017. Impact Factor: a crude yardstick which does not measure influence or impact on clinical practice. Spinal Cord, 55(9), pp.799-799. Diamandis, E.P., 2017. The Journal Impact Factor is under attack–use the CAPCI factor instead. BMC medicine, 15(1), pp.1-3.
  - *Citation removed.*

- Line 229: I would add here though how you found a positive relationship between APCs and impact factor. Why do you think there was a difference between the two? Could it be that when you have to pay for an article through an APC you pay more attention to impact factor, than when you do for a subscription-based journal that is paid through the University?
  - *We feel that the wording is clear enough, and that the reference to a future paragraph will be confusing. As such, we have left the wording as is, with only minor updates to reflect changes from the revised dataset.*

- Lines 234-236: This is the first time that PNAS is being used in the discussion of high profile journals as outliers to general trends. Would it be better to just focus on Nature and Science as you do throughout all of the Results / analyses?
  - *We have reworded for clarity in these references to journals, but we have reviewed the entire section for clarity of message, and include references to all outlier journals in the text corresponding to each of the figure panels.*

- Line 239. It's not clear how the citation of Neylon & Wu 2009 relates to your finding.
  - *We respectfully disagree. Neylon and Wu are discussing the differences between journal-level impact factors and article-level metrics, and how the two can and cannot be used to evaluate scholarship. They speak clearly to the question of how contrasts between the two perspectives need to be weighed. Seems an appropriate reference here.*

- Lines 253-256. Interesting to consider the work by Chua et al. 2017, who found a positive relationship between IF and citation rate for subscription-based journals, but a non-significant relationship between IF and citation rate for open access journals. Chua, S.K., Qureshi, A.M., Krishnan, V., Pai, D.R., Kamal, L.B., Gunasegaran, S., Afzal, M.Z., Ambawatta, L., Gan, J.Y., Kew, P.Y. and Winn, T., 2017. The impact factor of an open access journal does not contribute to an article's citations. F1000Research, 6.
  - *Now cited in the Discussion.*

- Lines 257-275: These two paragraphs are a bit too wordy. I would remove the two quotations from Garfield and DORA, as they do not add something novel to the discussion that is not covered by the rest of the text.
    - *That section has now been condensed and the quotations reduced to only one, from Garfield.*

- Lines 287-302. You could also discuss briefly your finding on editorships in relatively low-impact journals. Can you speculate why (perhaps the same motivations as for reviewing articles in said journals?)?
    - *Added text to clarify ideas on these topics.*

- Lines 287-302. You could embed here the monetary value of academics serving as reviewers… Aczel, B., Szaszi, B. and Holcombe, A.O., 2021. A billion-dollar donation: estimating the cost of researchers' time spent on peer review. Research Integrity and Peer Review, 6(1), pp.1-8.
    - *Added.*

- Discussion Point: It would have been interesting to touch upon why PLOS One is an outlier in terms of having a high APC but disproportionately many papers published. Could you speculate why that is?
    - *Some of these points are rather clear … PLoS One is a journal in which all EEB faculty can publish, simply because of its breadth. If we are talking about a journal like Ecology, only the ecologists in the department are likely to publish there. PLoS ONE, on the other hand, is there for everyone. Brief mention of these points is now added to the Discussion.*

- The authors might find the following technical briefing on open access patters in deep-sea biology published earlier this year, interesting. Available at: https://www.dosi-project.org/open-accesspublishing/ (link to pdf: https://www.dosi-project.org/wp-content/uploads/open-access-deepcommunity-briefing-1.pdf)
    - *Definitely interesting, but not added to the manuscript. There are many such reviews and overviews coming out these days.*

- Figure 3A. I would remove the circle and try to align the journal names somehow. It's a bit crowded but it should be possible. Right now, the first impression I get from looking this plot, is that something important is going on with these journals. Alternatively you could use journal abbreviations.
    - *The figures are now redone entirely, and the circle is removed. We believe that the labels on the figures are now much clearer.*

Reviewer: Ross Mounce
- I think this is really interesting work. I'm glad to see that knowledgeable KU librarians were brought into the collaboration to help with the work - similar research done by

academics-only such as https://peerj.com/articles/7850/ has faced criticism and doubts over validity for not including librarians… Key to this manuscript are the data, the software, and the exact processing/coding steps. I thank the authors for publicly archiving this in a suitable repository so they can be available for the reviewers and the public.

- *Thanks to the reviewer for this comment, and thanks for the detailed suggestions that follow. We have taken the reviewer's suggestions to heart, and decided as a result to re-derive the entire dataset, documenting each of the fields and each of the data entries more carefully. This has been time-consuming, but it has (we believe) made the dataset considerably more rigorous. As such, the entire analysis has been redone, including adding some new analyses that have become more relevant in the 2 years since we did the original analysis.*

● With regards to the underlying data https://kuscholarworks.ku.edu/handle/1808/32708 did the authors upload the correct file?
  - *Again, the dataset has been re-created completely… the current dataset linked in the manuscript is the correct one.*

● The data requires major revisions to make it robust, accurate, and clear.
  - *We have re-derived the dataset completely, starting with re-scraping Google Scholar for each of the KU EEB faculty members, and documenting each step carefully in all downstream steps.*

● In terms of coding "OA friendly" - I think a lot of journals have been miscoded, or else the coding methodology is insufficiently described to encapsulate what was done… Take for instance the journal Nature , the authors score this as '0' in terms of "OA friendly". From the manuscript I presume '0' translates to "unfriendly (i.e. author not allowed to post any post-review version of their paper)". Assigning '0' to Nature seems factually untrue to me. In actual fact, the journal Nature permits authors to make public their post-review accepted manuscript after only a 6-month embargo. See https://www.nature.com/nature-portfolio/editorial-policies/self-archiving-and-license-to-publish "When an article is accepted for publication in a Nature Portfolio journal via the subscription route, authors are permitted to self-archive the accepted manuscript on their own personal website and/or in their funder or institutional repositories, for public release six months after publication"

  Likewise, the journal Evolution, is given a '0' in the author's dataset, but it permits authors to upload their post-review accepted manuscript after a 12-month embargo (according to Sherpa/Romeo). I think the authors need to have a look again at how they've scored each of these journals for "OA-friendly" and perhaps even consider a more sophisticated coding scheme to encompass the variation in length of embargo and other OA-related attributes? The coding scheme of 0,1,2 is arbitrary and unusual, it needs more justification at the very least.
  - *We have taken a more nuanced approach to translating Sherpa/Romeo data on sharing policies for each of the journals. That is, instead of rating each journal on*

*a single scale, we have instead translated the Sherpa/Romeo information into a rating for the published, accepted, and submitted versions of the manuscript, as "yes", "no", "wait", "funder," and "$". These values do not capture all of the information in Sherpa/Romeo (e.g., an embargo of 6 months or of 5 years is "wait" in both cases), but it gets the major features without simply copying the full dataset, and we do not wish to create a separate version of the dataset, as the actual Sherpa/Romeo dataset is updated at regular intervals.*

- Why are there unfilled/blank cells and what do unfilled cells mean for each column?
  - *The blank cells indicated missing or unavailable data in the previous version. The present version is much more explicit—i.e., every value in every field is specified in the dataset key. Blank cells do persist because not all data items were available.*

- For instance, I don't list all <blank> cells, just emblematic examples which need to be addressed: a) the journal Molecular Ecology has a KU 2019 Subscription Price of <blank>. Molecular Ecology is NOT a fully OA journal, so the value in this cell ought to be >0 and not <blank>. If KU does not currently subscribe to Molecular Ecology then state that in the cell. Even then, I would have thought a representative price quote could still be sought and entered? Needs explanation at the very least!
  - *The KU subscription price information was clearly the most problematic of all of the data fields in the dataset. That is, we were constrained by "non-disclosure" clauses in some of the contracts, such that that information, in a few cases (including Molecular Ecology) simply could not be provided by the Libraries to us. We have included in the dataset what data COULD be shared with us by the Libraries, which is nevertheless a unique set of information. Regardless, as indicated above, now, all values in our dataset are defined rigorously.*

- b) Average (COUNTER) Usage of 2016 of PLOS ONE is given as <blank>. I think this ought to be coded more explicitly as e.g. UNKNOWN or DATA UNAVAILABLE rather than <blank>
- d) Why is there no COUNTER usage data for Phil Trans of the Royal Society B? It is another unexplained <blank> where I would expect there to be >0 data given this is a subscription/hybrid journal and the publisher does seem at first glance to provide COUNTER usage data.
  - *The counter usage data are very tricky. That is, they detect only the access events that come from someone using the KU-restricted library access page to get access to a journal. In that sense, an OA journal like PLoS ONE would not be included in the dataset. The Phil Trans absence is more complicated… in the newly derived version of the counter usage data, usage data for this journal were indeed included, but there are nevertheless still some data gaps in this field. They were, simply put, unavoidable, and we are making available in this dataset what information was available.*

- c) In their dataset, the journal Evolution has an Article Processing Charge value of <blank> . Evolution is a 'hybrid' journal that permits authors to have their article open access if they pay an APC, the 'standard' APC at 2022-05-17 is listed at a price of $3000 USD. <blank> insufficiently describes the situation of this journal in respect to providing open access &/or APC pricing. Please always be sure to refer to APCs as prices and not costs throughout the manuscript. The publisher often sets the APC as a price they determine, and this price may be completely divorced from the actual cost of running the publication venue.
    - *In our original dataset, we studiously ignored hybrid OA as a publishing mode, as we did (and still do) disagree with that approach on basic philosophical grounds. Nonetheless, in re-deriving the dataset for this study, we realized that the vast majority of the journals in our field have now added either an APC-driven gold OA option, or a hybrid OA option. As a consequence, we have now done a comprehensive sweep of journals to obtain the APC/hybrid OA fee information, which we feel adds an interesting new dimension to the study: the incredibly high "cost of openness" in ecology and evolutionary biology.*

- e) Why is "OA friendly" <blank> for many journals e.g. Phil Trans of the Royal Society B? This is just pure error - there must be a categorization for each and every journal given the current coding scheme.
    - *Yes, that was simply an inconsistency in our original dataset. We have now, as mentioned above, documented every possible value in the entire dataset, field by field, such that none of these inconsistencies should remain in the dataset.*

- f) Please provide an ISSN or ISSN-L for each journal, the journal name alone is not a unique identifier. "Journal of Insect Science" may refer to either one of the following two journals: Journal of Insect Science (Entomological Society of America) OR Journal of Insect Science (Indian Society for the Advancement of Insect Science)
    - *Very good point. We have added ISSN information for each journal in the analysis, save for a very few for which that information could not be found by us.*

- g) Not all open access journals are listed in DOAJ. Actually a great number of open access journals exist beyond DOAJ's listings e.g. read the The ==OA Diamond Journals Study - Science Europe==. This is more than just a theoretical problem. It is a practical one e.g. Bruce Liebermann has an article published in The Open Paleontology Journal (Bentham, now discontinued). This journal is/was fully open access. At one point in time The Open Paleontology Journal was DOAJ-listed, but the journal is no longer DOAJ-listed. Which brings me to another point: what version and/or timepoint of the DOAJ-listings did you use in this analysis? The DOAJ-listings are not static. It would be good to include some versioning information for reproducibility. If any journals were manually determined to be fully open access journals (outside of DOAJ-listings) perhaps state that? Pakistan Journal of Biological Sciences is one such title inc in the scope of this EEB analysis that to my eyes (looking at the journal website) seems openly accessible to readers, with articles distributed under CC-BY, yet it is not DOAJ-listed.

Oddly "Pakistan Journal of Biological Sciences" is also only listed as "1" for "OA-Friendly" which must be an error given the final pdfs are open online under CC-BY?

- ○ *We appreciate this comment. Indeed, the journal that the lead author of this paper edits, Biodiversity Informatics, was recently removed from DOAJ! We have now re-derived the dataset explicitly for December 2022 information held in DOAJ, so the information should be complete and up-to-date.*

  *The reviewer's point that DOAJ is not comprehensive is well-taken, and indeed the now-pervasive nature of hybrid OA access to journals makes things even more complicated. As such, we have re-cast how we treat "openness" in the dataset, avoiding making all-or-none classifications. To our knowledge, DOAJ is the most comprehensive source available on openness, so we have used this data source as the basis for our analyses.*

- I can't really trust this article until the underlying dataset is properly re-scored and more fully explained.
  - ○ *We have re-derived and documented each step in our processing in much more detail. We hope that the reviewer will now trust our summary entirely.*

- It would be nice if the authors wrote some nicer words about the value of research & scholarly writing that is NOT published in peer-reviewed journals. The way in which the 'not peer reviewed' or 'not published in a journal' were abruptly discarded from the analysis is troubling for the overall aim of the manuscript to look at what and where scholars and institutions are investing their time in supporting. Journals aren't the be all and end-all of scholarly communication. Monographs, book chapters, even blog posts can be excellent contributions to research, and all can be/are cited too.
  - ○ *We have added specification of our focus on peer-reviewed, journal-based publications to the "purpose paragraph." We have also included a short justification for this focus in the Methods.*

- What about selection bias? I note only 34/39 faculty were included and that the authors "requested they create [GS] profiles"
  - ○ *We appreciate the reviewer's concern, and provide additional information here, regarding our second iteration of the analyses. In this iteration, we were able to secure Google Scholar profiles for the following 35 faculty members: Folashade Agusto, Helen Miller Alexander, Brian Atkinson, K. Christopher Beard, James Bever, Sharon Billings, Justin Blumenstiel, Rafe Brown, Amy Burgin, Paulyn Cartwright, Jae Young Choi, Jocelyn Colella, Gerrit de Boer, Michael Engel, Bryan Foster, Jennifer Gleason, Richard Glor, Lena Hileman, Mark Holder, Kirsten Jensen, John K Kelly, Bruce Lieberman, Kelly Matsunaga, Robert Glen Moyle, Maria E. Orive, Townsend Peterson, Daniel Reuman, Andrew Short, Benjamin Sikes, Deborah Roan Smith, W. Leo Smith, Jorge Soberón, James Thorp III, Maggie Wagner, and James Walters. The only three faculty members who did not have GS profiles were Sara Baer, Mark Mort, and Raymond John Pierotti—of these individuals, Mort and Pierotti are not particularly*

*research-active. Because the great bulk of the department's productivity was included in our analyses, then, we are confident that no overt biases have entered our study.*

- I think you should discuss the weakness of this design a bit further. How many of the faculty specifically created a GS profile at the authors request? Why would an EEB faculty member actively not want a GS profile and may people like this (anti-Google?) have a different publication, editor, and review profile?
    - *We have added detail in the Methods about this. However, a few individuals either do not trust the Google Scholar platform, or did not wish to have the hassle of maintaining another scholar profile on another platform. We have inspected the recent publication history of the one research-active faculty member who did not create such a profile (Sara Baer), and did not see any marked differences between her publication behavior and that of the 35 faculty members who did have GS profiles. As such, we are confident that omission of these three faculty members' publication activity will not bias or affect negatively our analyses and interpretations.*

- What about the completeness of each of the 34 GS profiles included? No mention is made that GS profiles are malleable and owners of each profile can include/exclude publications on their GS profile at whim. I think you may want to consider evaluating this weakness, perhaps with reference to another data source?
    - *We have added text to the Methods to this effect, regarding the degree to which EEB faculty members were or were not editing or filtering their GS profiles. It appears that no members of the faculty were removing or biasing their profiles in any way that would affect the results of this study.*

- Likewise, what is the completeness of annual reports submitted to the department detailing editorial handlings and peer review reports? If a faculty member was in a rush to submit their annual report, might they not accidentally forget to include an editorship or a peer review done here or there? Perhaps you can cross-check with Publons for one or more faculty members to help assure the reader of the completeness of this data?
    - *Annual reports in KU EEB are the basis for advancement and merit salary increases. The preparation of annual reports is emphasized carefully by the Chair of the department. As such, we are confident that KU EEB faculty members are providing information that is reasonably complete. As to checking with Publons, several faculty members (including myself!!!!) indicated to us that they do not participate in Publons, in the belief that monetizing or "creditizing" peer review and editing activities may make it more likely that people will review papers that they are not qualified to review. Other faculty members had simply never heard of Publons! As such, we have not appealed to Publons, as it would be markedly less complete than the annual reports that we have used.*

- Please include a link to an archived page as close as possible (in time) to the time-point at which you measured the number and identity of EEB faculty at KU (1st March 2020). I found a snapshot of the page at the Internet Archive from December 2019 here: https://web.archive.org/web/20191206051920/http://eeb.ku.edu/faculty but if you have any other/better/closer archived snapshots to prove that assertion then please use them.
  - *We are concerned that the reviewer is doubting our honest about who was on the EEB faculty at the time of our analysis—we have no reason to change the representation of our colleagues in these analyses! We will hope that the reviewer is rather concerned about full replicability of our analyses, so we will pursue a much more positive, and usable, step: we are making our full dataset available to the community as part of this publication.*

- Line 158 "high-ranking" high-ranking according to who or what? I disagree with the concept of ranking journals (see various works by e.g. Bjorn Brembs). When I do rank journals e.g. in a spreadsheet I tend to do it alphabetically, so American Naturalist is 'high-ranking' for me :)
  - *This is a bit of over-interpretation of the wording that we used. We were talking about "summarizing the data by journal," so we were talking about "high-ranking" on the list of frequency of publication, and not in reference to any sort of quality ranking. We have reworded appropriately.*

- The analysis fails to acknowledge explicitly that lots of peer-reviewed journals have specific sections within them that are NOT peer-reviewed. How was this accounted for, if at all? Thus an author may have published an item in Nature, but that item might not necessarily be peer-reviewed. Were 'editorials' , 'reviews', 'letters of concern' , 'erratum notices' , 'corrigenda' and other such exotic article types all counted the same as typical 'primary research articles' ? It might be good to remark on the effect of this if so in a 'limitations' section of this manuscript.
  - *Now mentioned in both the Methods of the manuscript.*

- How representative is the EEB department at KU of any other EEB department at any other university? Some words to address this question might be nice?
  - *Brief mention and description now added to the end of the Introduction.*

- Please consider more clearly indicating that Clarivate's Journal Impact Factor (JIF)™ is a proprietary number. Specifically, capitalise all references to it AND include the ™ symbol to remind readers that it is also trademarked in addition to being untransparent, negotiable, statistically illiterate [1], irreproducible, and proprietary. Do not refer to merely 'impact factor(s)' – as such lower-casing might imply a generic, reproducible, statistically logical calculation. Also be more specific than just describing it as an 'average' (e.g. line 258). Clarivate famously calculates the mean rather than the median, which given the known highly skewed distribution of citations across articles in a journal is a statistically illiterate practice [1], hence you should tell readers it is the mean (which is statistically inappropriate) not the 'average'.

- ○ *We have qualified all mentions of impact factors throughout the manuscript. Where we will beg to differ with the reviewer is in the term average. We checked dictionaries, and "average" is defined as the sum of the observations divided by the number of observations, and is indeed synonymous with the less formal term "mean." As such, we have retained the more formal, and indeed correct, term "average.*

- Validity of the findings… The findings are not valid until they fix the dataset. The given dataset has a LOT of errors and unexplained cells.
  - ○ *As mentioned above, we have re-derived the entire data matrix, and have paid special attention to completeness and documentation. At the same time, we hope that the reviewer will understand that assembling such a multi-dimensional dataset is not simple, and that gaps will inevitably exist in the data matrix. We have done our utmost to (1) fill all data gaps, and (2) assure that no systematic biases enter into its content. We are—at least to our own satisfaction—confident that the data accurately approximate the publishing behavior of our department.*

Reviewer: Dominique Roche
- I was pleased to see that the authors archived the data and custom script used in this study. However, I would encourage the authors to also share the script used for the statistical analyses (test of relationships between variables) and to create the figures
  - ○ *We are sharing all scripts and all code that has been used in creating the results of this project, including creating the figures.*

- … improve the dataset by providing metadata (e.g., a comprehensive description of the column headings, units and abbreviations) in a separate readme file and applying tidy data principles (no blank cells, one value per cell [i.e., no units in cells]) to facilitate reuse (White et al. 2013).
  - ○ *As detailed above, all of the dataset has been re-done, to assure highest-quality information. We have now provided a readme file with the information requested.*

- Relationships between variables were examined by means of regression analysis, which assumes that one variable is dependent on the other. This dependency is not straightforward in almost all the relationships examined and, as such, the use of a correlative approach is more appropriate. The authors should consider reporting correlation coefficients rather than coefficients of determination.
  - ○ *We respectfully disagree. Correlation analyses are used when the question of interest is the degree to which two variables covary. In this case, we are interested in the degree to which one variable (e.g., JIF) depends on the (fixed) values of the other variable (e.g., journal subscription price). That is, we are coming to the analysis with very clear questions that are not about the degree to which two variables covary, but rather about the degree to which one variable responds to the other. In these cases, regression is the clear choice over correlation.*

- P-values should also be reported to three decimal places rather than above or below a critical value (Head et al. 2015).
  - *Again, we respectfully disagree with the reviewer's suggestion. While we appreciate the p-hacking arguments in the Head et al. (2015) paper that the reviewer cites, we point out that the really important number in thinking about a regression analysis is the $R^2$ value, known as the coefficient of determination, which tells the proportion of variation in the dependent variable that is explained by variation in the independent variable. We are focused on testing hypotheses, and when $P < 0.05$, the null hypothesis is rejected. Regardless, we provide the coefficient of determination, which is the crucial number.*

- The three paragraphs from L257-285 seem out of place and/or weekly linked to the text in this section of the manuscript. The content of these paragraphs is fine (although it could be condensed) but links to the discussion of the results are unclear. Consider rewriting, moving the text to another section, or drawing tighter links to the results discussed in the text of this section.
  - *We condensed the three paragraphs to two, and tied the text more tightly to the results of this paper.*

- Numerous references cited in the text do not appear in the bibliography.
  - *We believe that no references cited in the text were in the bibliography of the original manuscript. However, we have checked the entire manuscript, and all references cited in the text are in the bibliography in this version.*

- L55-64. Consider citing (Mekonnen et al. 2022) and (Larios et al. 2020).
  - *Now cited*

- L77. Consider defining 'journal use' on first use.
- L79. Comparator needed in this sentence.
  - *Both of these suggestions are appreciated, but are immaterial now, as we removed the relevant sentences in the interest of brevity overall in the manuscript.*

- L172-174. This sentence seems more appropriate for the discussion.
  - *Sentence removed entirely.*

- L185. Consider specifying EEB publications at the University of Kansas.
  - *The meaning of "EEB", and its location at the University of Kansas, is specified in the first paragraph of the Methods.*

- L232. Consider referring the reader to the relevant figure.
  - *Added.*

- Figure 3. The caption mentions publication frequency but the y axis in panel A is number of EEB papers. Please specify whether this is a frequency or a number and, in the latter case, the time period covered.
    - *It is a number. Now clarified in the figure caption.*

---

## Round 0.3 · accepted · Accept

Thank you for making the corrections and carefully addressing the (most helpful and attentive) referee's comments.